# Oxytocinergic projection from the hypothalamus to supramammillary nucleus drives recognition memory in mice

**Junpei Takahashi[1], Daisuke Yamada[1], Wakana Nagano[1], Yoshitake Sano[2], Teiichi Furuichi[2], Akiyoshi Saitoh[1]***

**1** Laboratory of Pharmacology, Faculty of Pharmaceutical Sciences, Tokyo University of Science, Chiba, Japan, **2** Department of Applied Biological Science, Faculty of Science and Technology, Tokyo University of Science, Chiba, Japan

* akiyoshi_saitoh@rs.tus.ac.jp

**Data Availability Statement:** Data are all contained within the Supporting Information files.

## Abstract

Oxytocin (OXT) neurons project to various brain regions and its receptor expression is widely distributed. Although it has been reported that OXT administration affects cognitive function, it is unclear how endogenous OXT plays roles in cognitive function. The present study examined the role of endogenous OXT in mice cognitive function. OXT neurons were specifically activated by OXT neuron-specific excitatory Designer Receptors Exclusively Activated by Designer Drug expression system and following administration of clozapine-N-oxide (CNO). Object recognition memory was assessed with the novel object recognition task (NORT). Moreover, we observed the expression of c-Fos via immunohistochemical staining to confirm neuronal activity. In NORT, the novel object exploration time percentage significantly increased in CNO-treated mice. CNO-treated mice showed a significant increase in the number of c-Fos-positive cells in the supramammillary nucleus (SuM). In addition, we found that the OXT-positive fibers from paraventricular hypothalamic nucleus (PVN) were identified in the SuM. Furthermore, mice injected locally with CNO into the SuM to activate OXTergic axons projecting from the PVN to the SuM showed significantly increased percentage time of novel object exploration. Taken together, we proposed that object recognition memory in mice could be modulated by OXT neurons in the PVN projecting to the SuM.

## Introduction

Oxytocin (OXT) is a peptide hormone that is synthesized in the paraventricular hypothalamic nucleus (PVN) and the supraoptic nucleus (SON). Projections of OXT neurons are widely distributed in the regions responsible for cognitive function. For example, previous reports discovered that OXT neurons projects to the hippocampus (Hip), perirhinal cortex, entorhinal cortex (Ent), and supramammillary nucleus (SuM) [1, 2]. Additionally, OXT receptors are reported to be expressed in the Hip, Ent, and SuM [1, 2]. The OXT receptor, a 7-transmembrane G protein-coupled receptor capable of binding to either $G_{i/o}$ or $G_{q/11}$ proteins, activates

**Funding:** This study was supported by a Grant-in-Aid for JSPS Fellows (Grant number JP 21J20036 to J.T.). https://www.jsps.go.jp/english/e-fellow/ YES -The funders had role in study design, data collection and analysis, decision to publish, or preparation of the manuscript.

**Competing interests:** The authors have declared that no competing interests exist.

a set of signaling cascades, such as the MAPK, PKC, PLC, CaMK, and CREB pathways [1, 3, 4]. The OXT receptors in the hippocampal dentate gyrus (DG) are mainly expressed on GABAergic neurons [5], whereas the OXT receptors in the CA1, CA2, and CA3 are expressed on glutamatergic neurons [5, 6]. Moreover, OXT has been shown to regulate social memory. Mice lacking OXT or OXT receptor gene in the CA2 area exhibited aberrant social memory [5, 7]. Furthermore, OXT neuron-specific deficiency of $Ca^{2+}$-dependent activator protein for secretion 2 (CAPS2), which regulates dense core vesicle exocytosis in OXT neurons, impairs social memory [8]. These reports suggest that endogenous OXT neurons play a pivotal role in regulating learning and memory functions, although the aspects regarding non-social memory are unclear.

Previous reports suggested that SuM, above the mammillary bodies, has an important role in learning and memory [9, 10]. Neurons in the SuM directly project to DG and CA2. SuM neurons regulate hippocampal theta rhythm [9, 11, 12]. The SuM-DG projections regulate new environmental memory, and the SuM-CA2 projections regulate social memory. Moreover, OXT receptors and OXTergic axons are localized in the SuM [12, 13]. *In vitro* electrophysiological experiments demonstrated that the application of OXT receptor agonist (Thr4,Gly7)-oxytocin increased the neuronal spikes of the SuM; these effects were antagonized when treated with the OXT receptor antagonist, (d(CH2)51,Tyr(Me)2,Thr4,Orn8,des-Gly-NH29)-Vasotocin [13]. It is unclear whether activation of endogenous OXTergic projections regulates OXT receptor-mediated neuronal activity in the SuM.

A previous study reported that intracerebroventricular administration of OXT enhanced long-term spatial and object memories [3, 14, 15]. Another study indicated that OXT perfusion in hippocampal slice enhanced the late phase of long-term potentiation (L-LTP) in electrophysical studies [3, 14]. Recently, we reported that OXT perfusion in hippocampal slice reversed Aβ-induced impairment of LTP [16]. Furthermore, intracerebroventricular and internal administration of OXT reversed Aβ-induced spatial memory impairment, as demonstrated by the undertaken Morris Water Maze (MWM) and Y-maze test [17]. With these results, we proposed that the administration of exogenous OXT in the central nervous system maintains or reinforces learning and memory. However, endogenous OXT's role in cognitive function remains unclear.

Although exogenous OXT has been reported to regulate learning and memory performance, studies clarifying whether endogenous OXT regulates learning and memory are limited. Therefore, the present study employed a chemogenetic approach to activate OXTergic neurons in the PVN specifically. Here, the OXTergic neurons activation, projecting from the PVN to the SuM, enhanced mice object recognition memory.

## Materials and methods

### Animals

The Institutional Animal Care and Use Committee at Tokyo University of Science approved all the experimental protocols; these were conducted following the National Institute of Health and Japan Neuroscience Society guidelines and ARRIVE guidelines (approval no. Y22014). Total of 51 mice (3- to 4-months-old male Oxt-iCre knock-in mice, C57BL/6-Oxt[tm1.1(Cre)Ksak]) [8] were used. All animals had free access to food and water in an animal room, with temperature (23˚C ± 1˚C) and relative humidity (45% ± 5%) and with a 12-h light-dark cycle (lights were automatically switched on at 8:00 am) and 3–5 mice per cage.

### Stereotaxic surgery

For surgery, Oxt-iCre mice were used. All mice were deeply anesthetized by intraperitoneal injection of a mixed solution of 0.75 mg/kg medetomidine (NIPPON ZENYAKU KOGYO

CO., Ltd. Fukushima, Japan), 4.0 mg/kg midazolam (Maruishi Pharmaceutical Co. Ltd, Osaka, Japan), and 5.0 mg/kg butorphanol (Meiji Animal Health Co., Ltd, Kumamoto, Japan) dissolved in saline. Then, mice were placed in a stereotaxic apparatus. After the surgery, the mice were administered atipamezole (0.75 mg/kg, i.p.: Kyoritsu Seiyaku Co., Tokyo, Japan) to reverse the sedation. Then, the mice were monitored for ≥5 days for recovery (normal eating, drinking, and defecation).

## Virus injection

OXT neurons were chemogenetically activated by Cre-dependent transfection of DREADD, hM3Dq. Male Oxt-iCre mice were injected with recombinant adeno-associated virus (hSyn-DIO-hM3D(Gq)-mCherry-AAV8; $1.25 \times 10^{13}$ virus molecules/ml; Addgene) into the PVN (anterior–posterior: −0.8 mm, mediolateral: ±0.3 mm, dorsoventral: −5.2 mm). A Hamilton 10-μl syringe with a 30-gauge blunt-end needle was inserted into the brain for virus delivery. The virus was bilaterally injected at a speed of 100nl/min for 200nL per site using a pump (UMC4, Florida, USA). To activate hM3Dq, mice were administered clozapine N-oxide (CNO, 1 mg/kg, i.p.: ENZ Enzo Life Sciences, Inc. BML-NS105-0025). CNO was dissolved in saline.

## Chemogenetic pathway-selective activation

A guide cannula (BRC bio research center, Ibaraki, Japan) was implanted in the SuM (anterior–posterior: −2.8 mm, mediolateral: ±0.3 mm, dorsoventral: −5.0 mm) for pathway-selective activation by hM3Dq. Guide cannulas were firmly fixed to the skull using dental cement. Finally, a dummy cannula (Plastics one, BRC bio research center, Ibaraki, Japan) was inserted in the guide cannula to avoid clogging. One week after recovery, 3 μM CNO (200 nL) was infused locally into the corresponding brain area at a speed of 300 nL/min through an internal cannula (Plastics one, BRC bio research center, Ibaraki, Japan) connected to the guide cannula. CNO concentration was determined based on previous studies [18–20].

## Immunohistochemistry

All mice were anesthetized by intraperitoneal injection of a mixed solution of medetomidine (0.75 mg/kg), midazolam (4.0 mg/kg), and butorphanol (5.0 mg/kg) dissolved in saline. Mice were perfused transcardially with PBS followed by 4% paraformaldehyde (PFA, Nacalai tesque Inc, Kyoto, Japan). Brains were dissected, postfixed in 4% PFA overnight, and subsequently immersed in 30% sucrose in PBS for cryoprotection. Following the fixation and cryoprotection, tissues were embedded in O.C.T compound (catalog #4583, Sakura-Finetek, Tokyo, Japan) and frozen at −80˚C. The brains were sectioned coronally and sagittally with a thickness of 50 μm. Brain sections were stored at −20˚C in cryoprotection solution (30% ethylene glycol, 25% glycerol in 1× PBS).

In IHC, stored brains were washed using PBST (0.2% Triton X-100 in PBS) three times and subsequently deactivated with 0.3% hydrogen peroxide in PBST for 5 min. Subsequently, sections were washed using PBST three times and blocked with blocking solution [3% BSA (catalog #001-000-162, Lot; 90246 Jackson ImmunoResearch, Pennsylvania, USA) in PBST] for 1 h. Then, the sections were incubated with primary antibody diluted with blocking solution for two overnights. After washing with PBST, the sections were incubated with secondary antibody [biotinylated α-rabbit IgG (catalog #BA-1100, vector, California, USA)] diluted with blocking solution for 90 min. Furthermore, sections were washed using PBST three times and incubated with AB solution (catalog #PK6100, vector, California, USA) for 60 min. Subsequently, sections were washed using PBST three times and then incubated with DAB (catalog

#SK4100, vector, California, USA) for six min. After washing with PBST three times, the sections were treated with hematoxylin (catalog# 8656, Sakura-finetek, Tokyo, Japan) for 5 min. Then, the sections were washed using PBST three times and attached to the glass slides. Finally, the sections were dipped in hexane and mounted with Eukitt (catalog# 6.00.01.0002.04.01.EN ORSAtec, Bobingen, Germany). NDP viewer (HAMAMATSU PHOTONICS, Shizuoka, Japan) was used to obtain the digital images.

For detecting c-Fos and mCherry, we used rabbit-c-Fos antibody (catalog # 226 003, Lot; 4–66, Synaptic Systems, Göttingen, Germany), rabbit-Ds Red antibody (catalog #632496, Clontech, Fitchburg, USA).

## Analysis of c-Fos and mCherry

A digital file containing a brain atlas was superimposed over each photomicrograph to specify the brain regions. The number of c-Fos-positive neurons in the CA1, CA2, CA3, Perirhinal cortex (PRh), Ent, and Cingulate/Retrosprenial cortex (Cg/RS) was counted manually by an observer blinded to the treatment conditions. Five to six sections were used to analyze c-Fos positive cells in hippocampal regions (CA1, CA2, CA3, and DG). Two to three sections were used to analyze c-Fos positive cells in the other brain regions (PVN, SuM, Ent, PRh and Cg/RS). We analyzed the dorsal portion of the Hip, because dorsal, but not ventral Hip, is reportedly involved in spatial memory and object memory [21–24]. We examined c-Fos positive neurons following NORT in this study. To analyze c-Fos counting, we referred to several reports [25–27]. These reports analyzed c-Fos positive neurons in several brain regions by using unpaired t-test. The distribution of OXTergic projections was identified as follows. OXT-iCre mice were injected with AAV8-hSyn-DIO-hM3Dq-mCherry in the PVN to specifically express the mCherry on OXTergic neurons. Then, the distribution of mCherry-positive neurons or fibers was detected by immunohistochemical staining against mCherry.

## Y-maze test

The spatial working memory was examined by measuring the spontaneous alternation behavior of the mice in the Y-maze test, as described previously with some modifications [17]. The maze was made of black acrylic board. Each arm was 40 cm long, 12 cm high, 3 cm wide at the bottom, 10 cm wide at the top, and converged at an equal angle. Each mouse was placed at the end of one arm and allowed to move freely through the maze during an 8 min session. The series of arm entries were recorded visually. Alternation was defined as successive entries into the three arms on overlapping triplet sets. The effect was calculated as the percent alternation following the formula: Alternation (%) = ((number of alternations)/(total number of arm entries– 2)) × 100 (%). The arms were wiped down with paper between sessions.

## Novel object recognition test (NORT)

The object memory was examined by measuring the exploring novel object behavior of the mice in the NORT, as described previously with some modifications [28]. NORT consists of three tasks: Habituation, training, and test. These tasks were performed in an open-field apparatus (30 × 30 × 30 cm). We used object A (cube) and object B (snowman); there were no differences in preference between the objects.

First, mice explored in open-field-apparatus for 5 min in habituation task. Twenty-four hours after the habituation task, mice explored in open-field-apparatus where objects A or B were placed for 10 min or until the total time spent exploring the objects to 20 s. We conducted counterbalance during training session. Half of the mice were presented with Object A and the other half with Object B. Then 72 h after the training task, mice explored in open-field-

apparatus where object A and B were placed for 10 min. Object recognition memory was assessed by measuring the exploration time of novel objects during the test. We counted the object exploration time in a blinded fashion. Object exploration time was referred as the time the mouse touches an object. Exploration time of novel object (%) = (time of exploration time of object B) / (total time of exploration time of objects) × 100 (%). Data was manually analyzed.

### Statistical analysis

Data were represented as mean ± SEM. The significance of differences in data was evaluated using unpaired *t*-test. Analyses were performed using Graphpad Prism (Graphpad Software Inc., San Diego, CA, USA); *P* values <0.05 were considered statistically significant.

## Results

### Projection of OXTergic neurons in PVN

We investigated the brain regions where receive projections of OXTergic neurons in the PVN. Mice injected with AAV8-hSyn-DIO-hM3Dq-mCherry in the PVN were perfused 4–6 weeks later and were analyzed expression of mCherry in OXTergic projection areas by immunohisto-chemical staining. mCherry-positive cells were identified in PVN (Fig 1A), and mCherry-positive fibers were identified in PVN (Fig 1A) and SuM (Fig 1B).

### The number of c-Fos positive cells in PVN after administration of CNO

We examined whether OXTergic neurons in Oxt-iCre mice with hM3Dq-AAV infection were activated by administering CNO. An overview of the experimental schedule is presented in (Fig 2A). We perfused mice 120 min after an administration of CNO (1mg/kg, i.p.) followed by neural activity analysis by immunostaining for c-Fos (Fig 2B and 2C). We counted the number of c-Fos positive cells in PVN. There were significantly more c-Fos positive cells in CNO treatment than saline treatment in PVN (Fig 2D; $t(7)$ = 4.331, $P$ = 0.0034, unpaired *t*-test, **$P < 0.01$).

(A) PVN

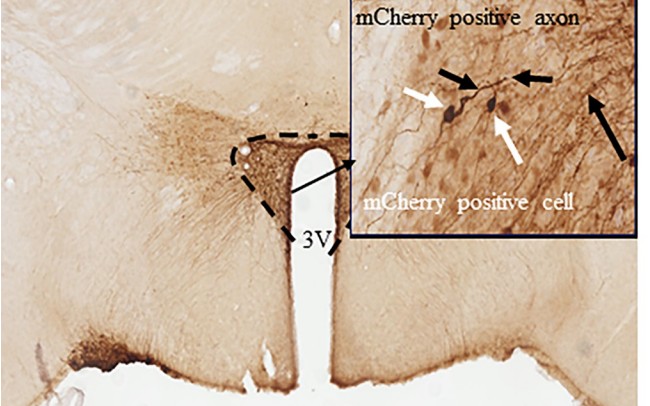

(B) SuM

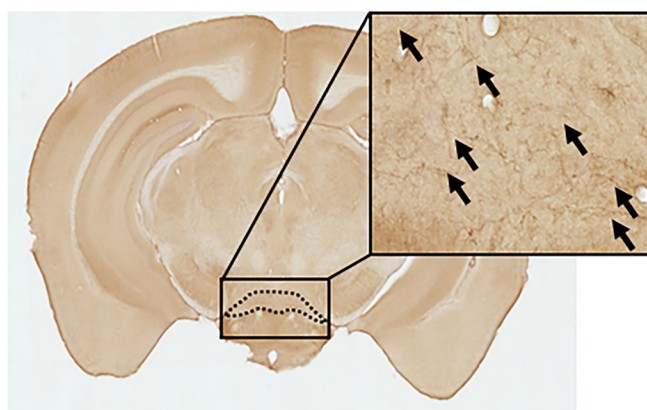

**Fig 1. Projection of OXTergic neurons in the PVN.** Immunohistochemical staining of mCherry 4–6 weeks after injecting AAV8-hSyn-DIO-hM3Dq-mCherry into the PVN. Image of the PVN (A) and SuM (B).

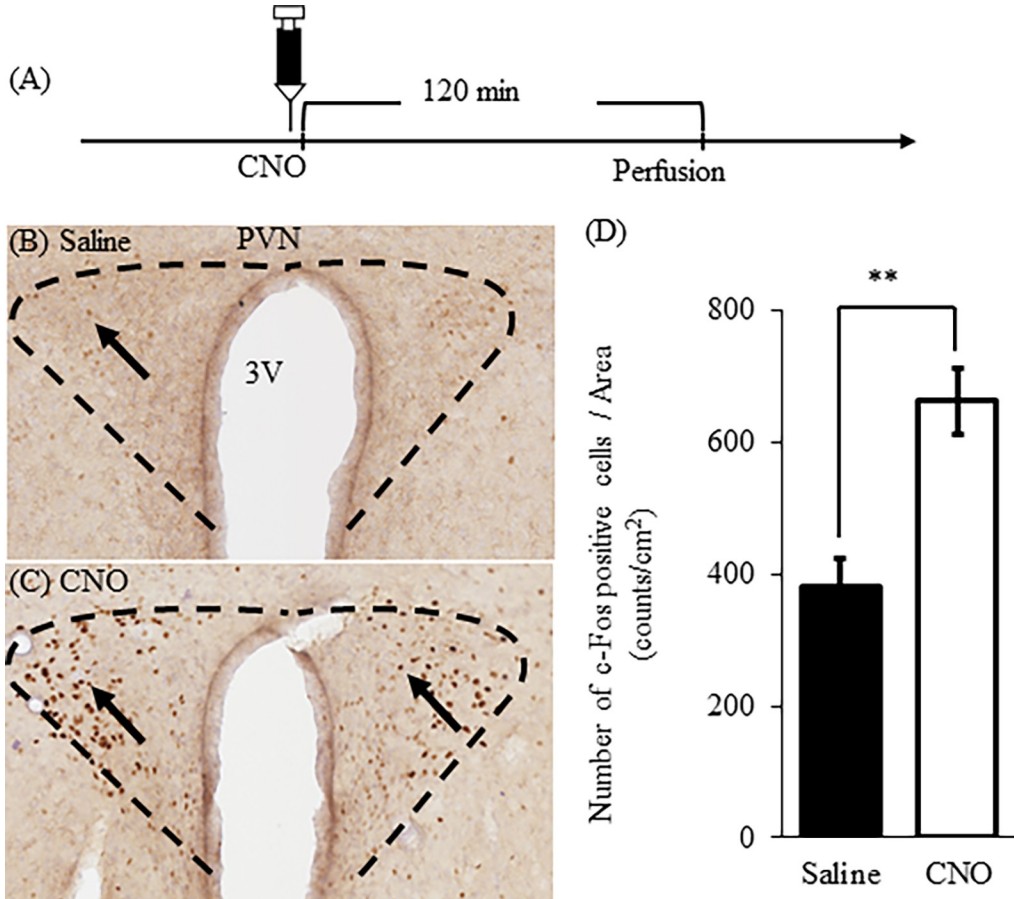

**Fig 2. The number of c-Fos positive cells in PVN after administration of CNO.** Immunohistochemical staining of c-Fos in the PVN 120 min after an intraperitoneal CNO administration (1.0 mg/kg). An overview of the experimental schedule is presented in (A). Image of PVN in saline-treated mice (B) and CNO-treated mice (C). The number of c-Fos positive cells in the saline and CNO (D). Data represents mean ± SEM (n = 4–5). Statistical analyzes were performed as unpaired t-tests. **$P < 0.01$.

## Effects of activation of OXTergic neurons in PVN on the Y-maze test and NORT

We examined whether activating OXTergic neurons affects learning and memory. Fig 3A presents the experimental schedule in Y-maze. The administration of CNO (1 mg/kg, i.p.) did not affect spontaneous alternation (Fig 3B; $t(13) = 0.1564$, $P = 0.8781$, unpaired $t$-test, ns: not significant) and total arm entries (Fig 3C; $t(13) = 1.533$, $P = 0.1493$, unpaired $t$-test, ns: not significant) in Y-maze. Fig 3D presents the experimental schedule in NORT. The CNO (1 mg/kg, i.p.) -treated mice did not affect the exploration time of the object during the training session than saline-treated mice (Fig 3E; $t(19) = 0.4889$, $P = 0.6305$, unpaired $t$-test, ns: not significant). The CNO treatment mice had more exploration time of novel object (%) during the test session than saline-treated mice (Fig 3F; $t(21) = 4.437$, $P = 0.0002$, unpaired $t$-test, ***$P < 0.001$).

## The number of c-Fos positive cells after NORT

We stained c-Fos 120 min after the NORT to examine which brain regions were involved in enhancing object recognition memory. An overview of the experimental schedule is presented in (Fig 4A). We counted the number of c-Fos positive cells in several brain regions. We found

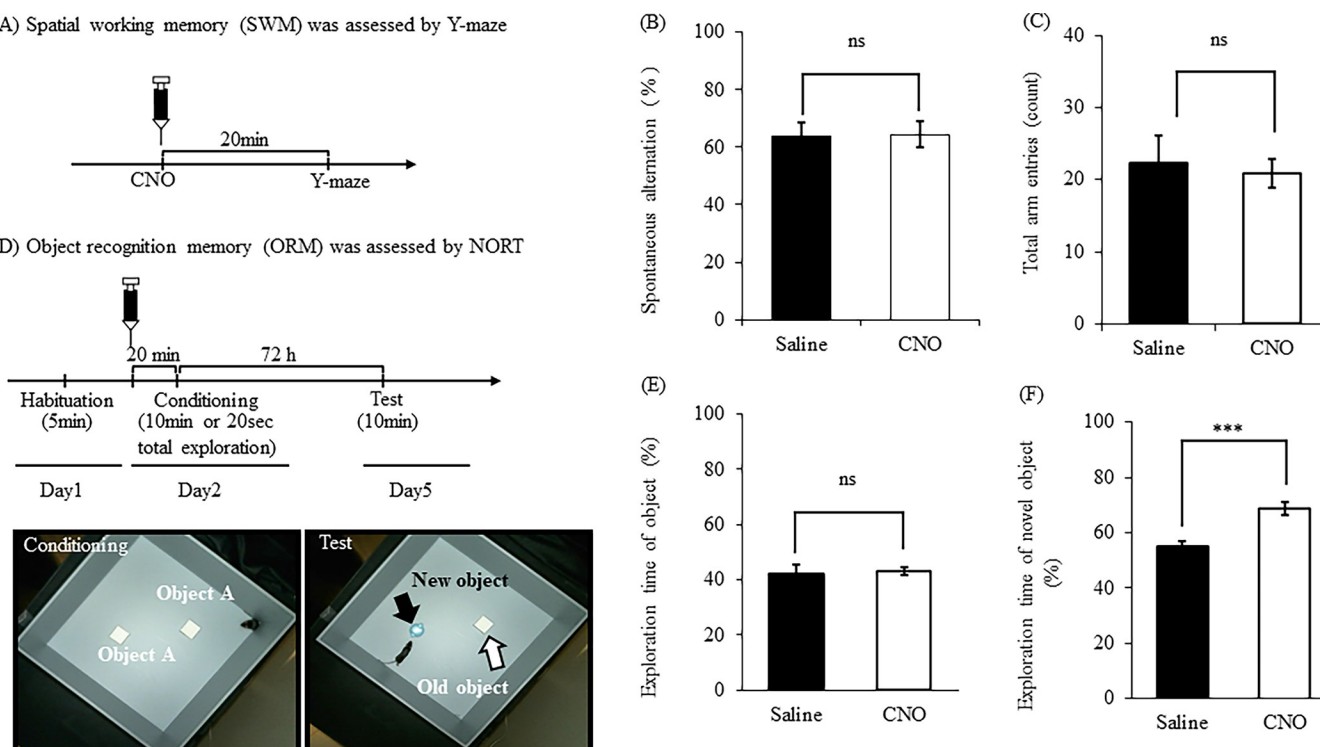

**Fig 3. Effects of activation of OXTergic neurons in PVN on the Y-maze test and NORT.** Mice received an intraperitoneal administration of saline or CNO (1 mg/kg) 20 min before Y-maze. An overview of the experimental schedule is presented in (A). Spontaneous alternation (B). Total arm entries (C). Mice received an intraperitoneal administration of saline or CNO (1 mg/kg) 120 min before the training session in NORT. An overview of the experimental schedule is presented in (D). Exploration time of object (%) during the training session (E). Exploration time of novel object (%) during the test session (F). Data were presented as the mean ± SEM (B, C; n = 7–8: E, F; n = 11–12). Statistical analyzes were performed as unpaired t-tests. ***$P < 0.001$, ns: no-significant.

that the number of c-Fos positive cells was significantly increased in CNO-treated mice compared to saline-treated mice in DG (Fig 4B, 4C and 4F; $t(12) = 2.645$, $P = 0.0214$, unpaired $t$-test, *$P < 0.05$) and SuM (Fig 4D, 4E and 4G; $t(12) = 2.265$, $P = 0.0428$, unpaired $t$-test, *$P < 0.05$). However, there was no difference between the number of c-Fos positive cells in CNO-treated mice compared to saline-treated mice in CA1, CA2, CA3, PRh, Ent and Cg/RS (Table 1; CA1, $t(5) = 0.4197$, $P = 0.6921$, unpaired $t$-test, ns: not significant); (Table 1; CA2, $t(5) = 2.068$, $P = 0.0935$, unpaired $t$-test, ns: not significant); (Table 1; CA3, $t(5) = 0.05772$, $P = 0.9562$, unpaired t-test, ns: no-significant);. (Table 1; PRh, $t(9) = 1.000$, $P = 0.3432$, unpaired $t$-test, ns: not significant); (Table 1; Ent, $t(9) = 0.5006$, $P = 0.6287$, unpaired $t$-test, ns: not significant); (Table 1; Cg/RS, $t(10) = 1.163$, $P = 0.2717$, unpaired $t$-test, ns: not significant).

## Effects of activation of OXTergic neurons in SuM by NORT

We implanted a guide cannula in SuM of mice to examine if OXTergic fibers in SuM were involved in enhancing object recognition memory. An overview of the experimental schedule is presented in (Fig 5A). Mice received a local injection of saline or CNO (3 μM, 200 nL) to SuM 20 min before the training session in NORT. The local injection of CNO did not affect exploration time of object during the training session compared to saline-treated mice (Fig 5B; $t(8) = 0.8773$, $P = 0.4059$, unpaired $t$-test, ns: not significant). CNO-treated mice exhibited a significant increase in exploration time of novel object (%) during the test session compared to saline-treated mice (Fig 5C; $t(8) = 2.419$, $P = 0.0419$, unpaired $t$-test, *$P < 0.05$).

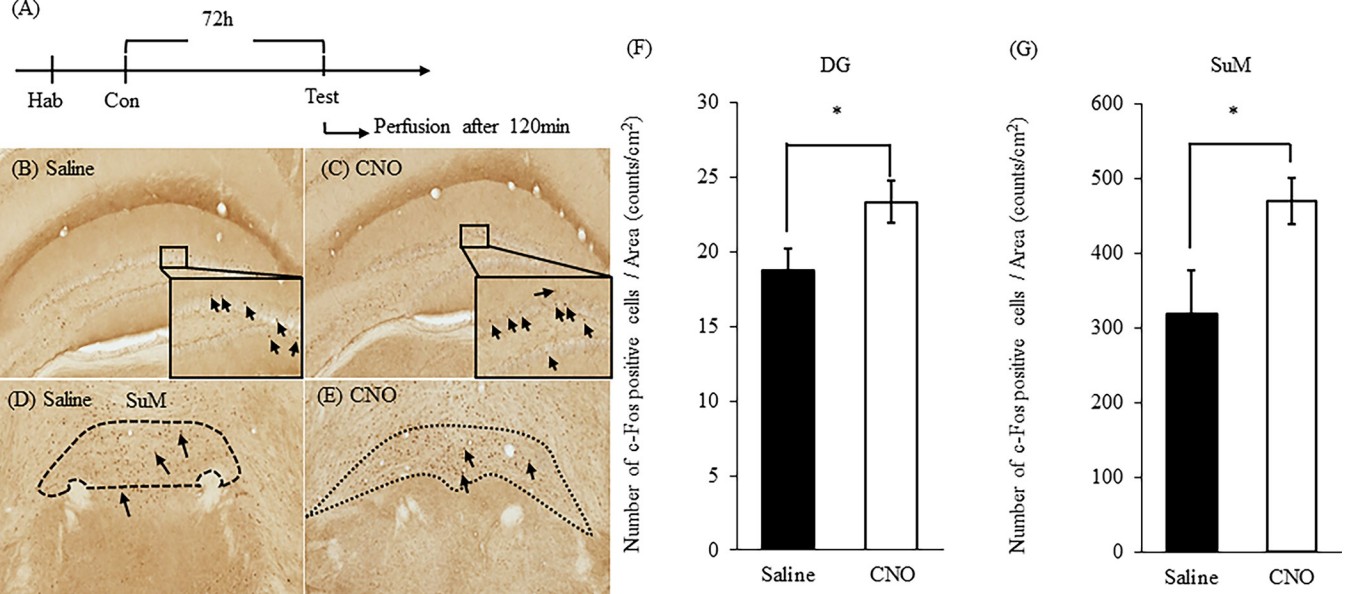

**Fig 4. The number of c-Fos positive cells after NORT.** Immunohistochemical staining of c-Fos 120 min after the NORT. An overview of the experimental schedule is presented in (A). Image of the DG in saline-treated mice (B). Image of the DG in CNO-treated mice (C). Image of the SuM in saline-treated mice (D). Image of the SuM in CNO-treated mice (E). The number of c-Fos positive cells in the DG (F) and SuM (G). Data were presented as the mean ± SEM (F; n = 7: E; n = 6–8). Statistical analyzes were performed as unpaired t-tests. *$P < 0.05$, ns: no-significant.

## Discussion

For the first time, this study found that endogenous OXT participates in object recognition memory via SuM. To investigate the projections of OXTergic neurons in PVN to different brain regions, we analyzed mCherry expression with ABC staining methods. Results show that mCherry-positive cells observed in PVN and mCherry-positive fibers were observed in SuM. These results suggested that OXTergic neurons in PVN projected to SuM, similar to previous reports [1, 2]. However, mCherry-positive fibers were not observed in Hip (include in CA1, CA2, CA3 and DG). It was reported that alkaline phosphatase staining did not detect OXT neurons projection in the Hip by using Oxt [cre/+]; Z/AP mice [29]. It is possible that the areas where OXT neuron projections could be detected depend on the mouse type and staining methods used.

CNO treatment increased the number of c-Fos positive neurons in the PVN. To ascertain that the identified post-treatment c-Fos positive neurons were indeed OXTergic neurons, we detected c-Fos and OXT neurons by immunofluorescence staining. We found that the double-

**Table 1. Expressions of c-Fos in the various brain areas after NORT.** Immunohistochemical staining of c-Fos 120 min after the NORT. Table showed the number of c-Fos positive cells in the PRh, Ent, CA1, CA2, CA3 and Cg/RS. Data were presented as the mean ± SEM. Statistical analyzes were performed as unpaired t-tests.

| Area | Saline (n) | CNO(n) | P value |
|---|---|---|---|
| PRh (perirhinal cortex) | 53±21(5) | 82±15(5) | 0.30 |
| Ent (entorhinal cortex) | 74±18(5) | 76±8.8(5) | 0.90 |
| CA1 | 3.8±0.3(3) | 3.4±0.9(4) | 0.70 |
| CA2 | 1.2±0.15(3) | 0.8±0.12(4) | 0.09 |
| CA3 | 13±3.8(3) | 13±0.9(4) | 0.96 |
| Cg/RS (cingulate/retrosplenial cortex) | 100±24(5) | 143±25(7) | 0.27 |

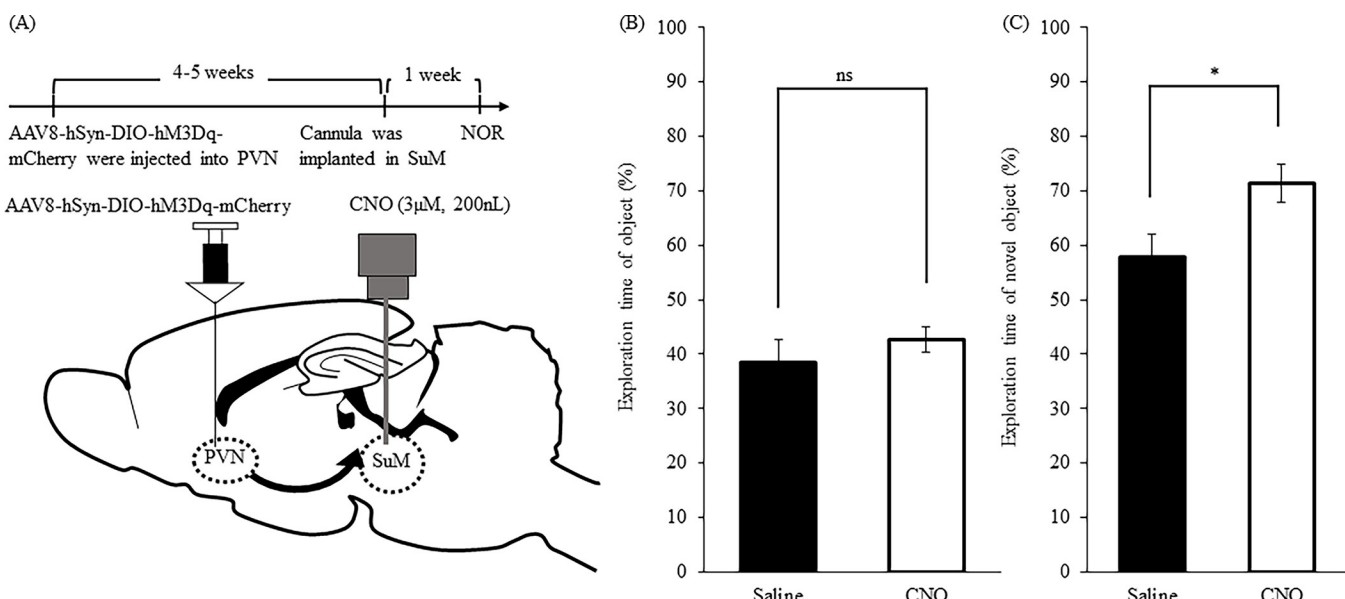

**Fig 5. Effects of activation of OXTergic neurons in SuM during the NORT.** Mice received a local injection of saline or CNO (3μM, 200nL) to SuM 20 min before the training session in NORT. An overview of the experimental schedule is presented in (A). Exploration time of object (%) during the training session (B). Exploration time of novel object (%) during the test session (C). Data were presented as the mean ± SEM (B, C; n = 5). Statistical analyzes were performed as unpaired t-tests. *$P < 0.05$, ns: no-significant.

labeled neurons (c-Fos+ and OXT+ neuron) were detected in PVN of CNO-treated mice (S3 Fig). These results suggested that c-Fos positive neurons were OXTergic neurons.

There was no difference in spontaneous alternation in the Y-maze test between saline- and CNO-treated mice, suggesting that endogenous OXT does not affect short-term spatial memory. In NORT, naïve mice had approximately 70% of the exploration time to novel object after 6 h of the training task (S1 Fig). Therefore, naïve mice could maintain object recognition memory after 6 h of training task. However saline-treated mice exhibited about 50% of the exploration time to novel object after 72 h of the training task, indicating that normal mice could not maintain object recognition memory after 72 h of the training task. Interestingly, CNO-treated mice exhibited about 70% of the exploration time to novel object during NORT, suggesting that CNO-treated mice could distinguish a new object from an old object. Thus, we propose that endogenous OXT enhances long-term object recognition memory. Our results are compatible with previous study in which OXT improved hippocampal L-LTP, which is important to maintain memory, but not early-phase LTP [3, 14]. Moreover, it has been reported that OXT enhanced long-term, but not short-term, spatial memory in eight-arm radial maze task in rodents [3]. Thus, previous studies and our findings suggest that OXT regulates cognitive functions in long-term, but not short-term, memory.

Following NORT, CNO treatment upregulated the number of c-Fos positive neurons in the SuM and DG. These results suggested possibility that activating OXTergic neurons enhanced the maintenance of long-term memory in mice via neurons in DG and SuM. Several reports suggested that neurons in DG and SuM are involved in object recognition memory. Ex vivo electrophysiological experiment demonstrated that the ratio of membrane currents mediated by AMPA receptor to NMDA receptor (AMPA/NMDA ratio), an index of neuronal plasticity, was increased in DG granule cells after NORT [30]. Moreover, bilateral lesions of dorsal DG in rats showed impaired discrimination of the two objects compared to the sham-operated rats [31], suggesting that dorsal hippocampal DG neurons modulate long-term object recognition

memory. Additionally, selective activation of axons of SuM projection neurons in the medial prefrontal cortex improved impairment of object recognition memory in 5×FAD mice [32], suggesting that SuM is also involved in object recognition memory. Thus, we propose that activating DG and SuM neurons enhance object recognition memory by activating OXT neurons.

Finally, chemogenetic pathway-selective activation of OXTergic axons in SuM increased the exploration time to novel object, suggesting that OXTergic axons in SuM projecting from PVN modulate object recognition memory. Neurons in the SuM directly project to DG and CA2. They are also involved in learning and memory [9, 12]. Thus, this study's findings indicate that OXTergic axons in SuM modulate object recognition memory via DG. The SuM-DG pathway modulates new environmental memory [9, 10]. Chen et al. injected AAV-DIO--ChR2-eYEP or AAV-DIO-eNpHR-eYEP into the SuM in SuM-Cre mice to manipulate the selective SuM-DG pathway [10]. The SuM-DG pathway, activated or inhibited with optogenetics, showed altered exploration behavior in contextual novelty environment [10]. Li et al. injected AAV-DIO-hM3Dq-mCherry or AAV-DIO-hM4Di-mCherry into SuM in mice. This study reported that CNO treatment increased c-Fos positive neurons in the DG of hM3Dq expressing mice, while CNO treatment decreased c-Fos positive neurons in the DG of hM4Di expressing mice. In addition, Li et al. injected AAV-DIO-Ch2R-mcherry and AAV-shVglut2 (known as a marker of glutamatergic neurons) into SuM in mice. Activating DG with optogenetics decreased c-Fos positive neurons in the DG of deletion of Vglut2 mice. Therefore, these results suggested that glutamatergic neurons in the SuM activate DG. Moreover, Li et al. assessed cognitive behavior by novel place recognition test and NORT [9]. CNO-treated mice could distinguish new place from old place by novel place recognition test. However, they could not distinguish a new object from an old object by NORT. Thus, contrary to our results, these findings suggested that glutamatergic neurons in SuM modulate spatial memory but not recognition memory [9]. This difference in the findings could be due to the difference in the time course of experiments. Li et al. conducted the test task 1 h after the training task, whereas we conducted the test task 72 h after the training task in NORT. These differences in hours after training (72 hours (long-term memory) vs. 1 hour (short-term memory)) may suggest that SuM neurons modulate the type of memory and time-dependent memory. Taken together, we proposed that OXTergic neurons in SuM projecting from PVN could enhance object memory via the activating of DG projecting from glutamatergic neurons in SuM.

Recent study reported that OXTergic axons in SuM modulate social memory via CA2 [33]. Keerthi et al. reported that the selective inhibition pathway of OXTergic axons in SuM decreased the investigation time to the novel rodents in social memory test [33]. Additionally, they found that the microinjection of OXT receptor antagonist in SuM decreased the investigation time to the novel rodents. Moreover, this report exhibited that 61% of vglut2 positive neurons in the SuM co-expressed OXT receptors with *in situ* fluorescent hybridization techniques. This indicates that OXTergic axons projecting to SuM from PVN modulates social memory via OXT receptors, which are predominantly expressed in the glutamatergic neuron in the SuM. Moreover, Keerthi et al. proposed that the PVN-SuM-CA2 pathway may be important for social memory, although they reported that it is unclear whether glutamatergic neurons with OXT receptors in SuM project to the hippocampal CA2 region [33]. Based on these reports, OXTergic axons in SuM modulate the SuM neurons projecting to DG; this neuronal participation enhances object recognition memory in NORT.

There are several reports demonstrating the modulatory effect of OXT on cognitive function. For example, we recently reported that OXT recovered impairment of spatial memory in Alzheimer's disease (AD) model mice established by intracerebroventricular injection of β-amyloid protein (Aβ), commonly used to mimic the pathogenesis of AD [17]. Furthermore, OXT recovered $A\beta_{25-35}$-induced impairment of hippocampal long-term potentiation [16].

Other researchers also reported the therapeutic effects of OXT in AD-like symptoms in animal studies. Additionally, intranasal OXT attenuated aluminum chloride (AL)-induced impairment of cognitive function, decreased AL-induced elevation of $A\beta_{1-42}$, and Tau levels [34]. Interestingly, intravenous injection of OXT nanogel reversed spatial memory deficits of APP/PS1 mice in MWM [35]. Moreover, AD patients demonstrated a decrease in plasma OXT concentration compared to healthy individuals [36]. Although we cannot examine the direct relationship between AD and OXTergic neuron activity in this study, our result suggested that OXTergic neuron activity may be involved in each stage of AD pathological progression.

Female mice may exhibit differences since the distribution of OXT receptors and OXT projection areas differ between female and male mice [2]. However, we did not use female Oxt-iCre mice to avoid the possibility of recombination in all cells in a Cre expressing fertilized egg. Moreover, DAT-Cre KI mice show sex-dependent changes in amphetamine-induced locomotor activity [37]. This report suggested that the phenotypes of Cre-KI mice were sex-dependent. Thus, we used male Oxt-iCre mice in this study.

## Conclusion

This study found that OXTergic projection from PVN to SuM drives recognition memory in mice for the first time. This study could be beneficial for future investigation of the role of physiological OXT in AD.

## Supporting information

**S1 Fig. The exploration preference to object in our laboratory conditions in NORT.** Figure showed the exploration time of object (%) during the training session and the test session. Data was presented as the mean ± SEM. (n = 8). Statistical analyzes was performed as unpaired t-tests. *P < 0.05.
(TIF)

**S2 Fig. The difference exploration time of object during the training session and the test session in NORT.** Figure showed the difference exploration time of object (s) during the training session and the test session. Test session was conducted 72 h after the training session Data was presented as the mean ± SEM. (n = 12). Statistical analyzes was performed as unpaired t-tests. **P < 0.001.
(TIF)

**S3 Fig. The immunohistochemical staining of c-Fos and OXT neurons.** Immunohistochemical staining of c-Fos and OXT neurons 120 min after administration of CNO. Magnification of 20x image of the PVN in saline-treated mice (A) and CNO-treated mice (B). Magnification of a 40X image of the PVN in saline-treated mice (C) and CNO-treated mice (D). Arrow indicate the double-labeled neurons (c-Fos + and OXT+).
(TIF)

**S1 Raw data.**
(XLSX)

**S2 Raw data.**
(XLSX)

**S3 Raw data.**
(XLSX)

**S4 Raw data.**
(XLSX)

## Acknowledgments

We would like to thank Enago (www.enago.jp) for the English language review.

## Author Contributions

**Conceptualization:** Daisuke Yamada, Yoshitake Sano, Akiyoshi Saitoh.

**Data curation:** Junpei Takahashi, Wakana Nagano.

**Formal analysis:** Junpei Takahashi.

**Funding acquisition:** Junpei Takahashi.

**Investigation:** Junpei Takahashi, Wakana Nagano.

**Project administration:** Akiyoshi Saitoh.

**Supervision:** Daisuke Yamada, Yoshitake Sano, Teiichi Furuichi, Akiyoshi Saitoh.

**Writing – original draft:** Junpei Takahashi.

**Writing – review & editing:** Daisuke Yamada, Yoshitake Sano, Teiichi Furuichi, Akiyoshi Saitoh.

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
