## [Decision Letter · Decision Letter 0]

25 Aug 2023

PONE-D-23-22233Oxytocinergic Projection from the Hypothalamus to Supramammillary Nucleus Drives Recognition Memory in MicePLOS ONE

Dear Dr. Saitoh,

Thank you for submitting your manuscript to PLOS ONE. After careful consideration, we feel that it has merit but does not fully meet PLOS ONE’s publication criteria as it currently stands. Therefore, we invite you to submit a revised version of the manuscript that addresses the points raised during the review process.

We look forward to receiving your revised manuscript.

Kind regards,

Peng Zhong, Ph.D.

Academic Editor

PLOS ONE

3. Please include your tables as part of your main manuscript and remove the individual files. Please note that supplementary tables (should remain/ be uploaded) as separate "supporting information" files.

Additional Editor Comments:

The findings are interesting. However, please provide the more convincing image demonstrating the projections of SuM. The reviewers' comments are needed to be addressed carefully, especially ones from Reviewer 1.

Reviewers' comments:

Reviewer's Responses to Questions

**Comments to the Author**

1. Is the manuscript technically sound, and do the data support the conclusions?

Reviewer #1: Yes

Reviewer #2: Partly

2. Has the statistical analysis been performed appropriately and rigorously? 

Reviewer #1: Yes

Reviewer #2: Yes

3. Have the authors made all data underlying the findings in their manuscript fully available?

Reviewer #1: Yes

Reviewer #2: Yes

4. Is the manuscript presented in an intelligible fashion and written in standard English?

Reviewer #1: No

Reviewer #2: Yes

5. Review Comments to the Author

Reviewer #1: This manuscript describes results of experiments testing the influence of oxytocin projection neurons from periventricular nucleus (PVN) to the supramammillary nucleus (SuM) on long-term memory for objects. The excitatory DREADDs receptor, hM3Di was expressed in oxytocin neurons of the PVN, mice were subsequently administered systemic CNO and tested for the influence on c-Fos expression in SuM, spatial working in a Y-maze task, and object recognition memory. Systemic CNO enhanced the preference of the mice to explore the novel object during a test session conducted 72 h after the initial training session. A follow-up study demonstrated a similar enhancing effect after mice received local infusion of CNO into the SuM. The authors conclude that oxytocin projection neurons of the PVN play a significant role in object recognition memory. There is compelling evidence from prior reports that oxytocin influences social memory, and converging evidence has defined circuits supporting this effect. The present manuscript expands on this literature by demonstrating the involvement of oxytocin in a different form of recognition memory. The results are interesting and the breadth of the approach here is commendable. There are some issues outlined below that should be addressed.

1. The details of how the object recognition task was conducted are missing. (A) For example, page 7 (lines 174-175) indicates that object A and B were used, and line 177 indicates that two object A's were presented during the training session. The object recognition task is based on spontaneous responses of the mice, and one must be careful to counterbalance the objects presented. For example, it would be important for the training session to present two object A's to half of the mice, and two object B's to the other half of the mice. The authors state that there were no differences in preference between the objects; however, counterbalancing objects presented during training and testing brings rigor to the approach.

(B) The measure used to assess memory was preference ratio, which suffers from a limited range: chance performance in 50%, and substantial preference for the novel object is rarely more than 75%. A better measure of test session memory performance is the Discrimination Ratio, calculated by taking the difference score (time exploring object B - time exploring object A) and dividing the result by the total time spent exploring both objects. Discrimination ratio scores better account for overall differences in total object exploration between individual mice, and the data tend to have a wider range of scores than that of preference ratio. The discrimination ratio is a well accepted and more rigorous measure of object memory than preference ratio. The authors are encouraged to present their test session object recognition data as discrimination ratio scores.

(C) Page 7, line 182 states that the "Data was manually analyzed." The authors should provide more detail as to what this means. How was object exploration scored, what constituted object exploration, and who scored the data? How was scientific rigor upheld while manual analysis was occurring?

2. Details are missing as to how the c-Fos counts were conducted and analyzed. How many sections from each mouse were counted for each of the regions (CA1, CA2, CA3, ... etc.)? Were the counts conducted blind? c-Fos counts were normalized by the area of the respective region. However, how was this determined given the need to control for the thickness of tissue and the differences in the size of the respective structures across sections? Also, background varies across a piece of tissue and across slides, so what approach was taken to achieve consistency across slides?

3. The c-Fos data were analyzed by multiple t-tests - that is, an unpaired t-test (saline- vs. CNO-treated) for normalized c-Fos counts in each region. The authors should consider that conducting multiple t-tests carries the risk of yielding a significant outcome by chance. Given that the data were normalized (though see comment #2 above), why not instead conduct a two-way RMANOVA (treatment group as between-subjects variable, and region of interest as the repeated measures/within-subjects variable)?

4. The manuscript refers to Table 1, but this information was not provided.

Minor issues

a) page 6, line 153: the statement that "...dorsal, but not ventral Hipp[ocampus] is involved in cognitive function..." is not well supported by the literature. For example see: Jin & Maren (2015) Scientific Reports; Okuyama et al. (2016) Science; Contreras et al. (2018) Hippocampus; Rao et al, (2019) Cell Reports; Jimenez et al. (2020) Nature Communications; to name a few example references.

b) the manuscript contains an overwhelming number of acronyms (OXT, ORM, SuM, PVN, ABC, CNO, NORT, AD, etc.), which makes some of the sentences challenging to understand. This reviewer suggests limiting the acronyms to a minimal number of terms that are used throughout.

c) page 10, line 249-252: This sentence is quite difficult to understand. Please revise.

d) page 11, line 267: "competed" should be revised to "compared"

e) page 13, line 314-315: The statement "...our result suggested that OXTergic neuron activity may be involved in each stage of AD pathological progression." is not supported by the results described in the current manuscript.

Reviewer #2: 1. As stated by the author in the introduction, "Projections of oxytocin (OXT) neurons exhibit extensive distribution across regions associated with cognitive functions. For instance, prior studies have revealed that OXT neurons project to the hippocampus (Hip), perirhinal cortex, entorhinal cortex (Ent), and supramammillary nucleus (SuM)." However, the findings from immunohistochemical staining (Fig. 1) merely illustrate the fibers within the paraventricular nucleus (PVN) and supramammillary nucleus (SuM). It would be advantageous to incorporate outcomes from other cerebral regions, notably the hippocampus (DG).

2. Regarding c-Fos staining and quantification subsequent to CNO treatment, it is advisable for the authors to undertake additional verification to ascertain that the identified c-Fos positive neurons post-treatment indeed are OXTergic neurons.

3. Does the localized administration of CNO to the SuM have an impact on the DG? While the results section does not explicitly address the SuM-DG pathway, it extensively delves into this topic in the discussion. It would be better for the authors to substantiate their conceptual arguments in the discussion by presenting relevant empirical findings.

4. In the discussion section (page 11, line 270), the authors put forth the assertion that “Thus, we propose that activating DG and SuM neurons enhance ORM by activating OXT neurons.” It would be judicious for the authors to complement this proposal by providing evidential support. Specifically, they could present data that demonstrate the comparable effects observed in “Effects of activation of OXTergic neurons in SuM by NORT”, where local activation of oxytocinergic fibers in the DG produced similar outcomes.

5. In this study, only male mice were used as the subjects. Could potential distinctions arise in female mice?

6. PLOS authors have the option to publish the peer review history of their article (what does this mean?). If published, this will include your full peer review and any attached files.

Reviewer #1: **Yes: **Robert W. Stackman Jr.

Reviewer #2: No

---

## [Author Response · Author response to Decision Letter 0]

26 Sep 2023

Responses to the comments raised by Reviewer #1

Reviewer #1 comment

This manuscript describes results of experiments testing the influence of oxytocin projection neurons from periventricular nucleus (PVN) to the supramammillary nucleus (SuM) on long-term memory for objects. The excitatory DREADDs receptor, hM3Di was expressed in oxytocin neurons of the PVN, mice were subsequently administered systemic CNO and tested for the influence on c-Fos expression in SuM, spatial working in a Y-maze task, and object recognition memory. Systemic CNO enhanced the preference of the mice to explore the novel object during a test session conducted 72 h after the initial training session. A follow-up study demonstrated a similar enhancing effect after mice received local infusion of CNO into the SuM. The authors conclude that oxytocin projection neurons of the PVN play a significant role in object recognition memory. There is compelling evidence from prior reports that oxytocin influences social memory, and converging evidence has defined circuits supporting this effect. The present manuscript expands on this literature by demonstrating the involvement of oxytocin in a different form of recognition memory. The results are interesting and the breadth of the approach here is commendable. There are some issues outlined below that should be addressed.

Reviewer #1 comment 1

The details of how the object recognition task was conducted are missing. (A) For example, page 7 (lines 174-175) indicates that object A and B were used, and line 177 indicates that two object A's were presented during the training session. The object recognition task is based on spontaneous responses of the mice, and one must be careful to counterbalance the objects presented. For example, it would be important for the training session to present two object A's to half of the mice, and two object B's to the other half of the mice. The authors state that there were no differences in preference between the objects; however, counterbalancing objects presented during training and testing brings rigor to the approach.

(B) The measure used to assess memory was preference ratio, which suffers from a limited range: chance performance in 50%, and substantial preference for the novel object is rarely more than 75%. A better measure of test session memory performance is the Discrimination Ratio, calculated by taking the difference score (time exploring object B - time exploring object A) and dividing the result by the total time spent exploring both objects. Discrimination ratio scores better account for overall differences in total object exploration between individual mice, and the data tend to have a wider range of scores than that of preference ratio. The discrimination ratio is a well accepted and more rigorous measure of object memory than preference ratio. The authors are encouraged to present their test session object recognition data as discrimination ratio scores.

(C) Page 7, line 182 states that the "Data was manually analyzed." The authors should provide more detail as to what this means. How was object exploration scored, what constituted object exploration, and who scored the data? How was scientific rigor upheld while manual analysis was occurring?

Response to comment 1

Thank you for your helpful suggestions. (A) We conducted counterbalance during the training session. Half of the mice were presented with Object A and the other half with Object B. We have now added this information to the Materials & Methods section (line 187). (B) We analyzed the difference score (time exploring new object – time exploring old object). These results were presented in supplement figure 2. (C) We counted the object exploration time in a blinded fashion. Object exploration time is referred to the time during which the mouse touches an object. We added this information to the Materials & Methods (line 190).

Reviewer #1 comment 2

Details are missing as to how the c-Fos counts were conducted and analyzed. How many sections from each mouse were counted for each of the regions (CA1, CA2, CA3, ... etc.)? Were the counts conducted blind? c-Fos counts were normalized by the area of the respective region. However, how was this determined given the need to control for the thickness of tissue and the differences in the size of the respective structures across sections? Also, background varies across a piece of tissue and across slides, so what approach was taken to achieve consistency across slides?

Response to comment 2

Thank you for your helpful suggestions. We conducted c-Fos counts blindly. Five to six sections were used to analyze c-Fos positive cells in hippocampal regions (CA1, CA2, CA3 and DG). Two to three sections were used to analyze c-Fos positive cells in other brain regions (PVN, SuM, Ent, PRh, and Cg/RS). We added relevant text responding to these comments in the Materials & Methods section (line 156). Furthermore, slices were cut using a cryostat, which can perfectly regulate section thickness. We calculated the area of the respective regions by NDP views. We think that size variation can be minimized by measuring several slices. Finally, we used the same reaction time for antibody incubation, AB reaction, and DAB staining to achieve a consistent background level.

Reviewer #1 comment 3

The c-Fos data were analyzed by multiple t-tests - that is, an unpaired t-test (saline- vs. CNO-treated) for normalized c-Fos counts in each region. The authors should consider that conducting multiple t-tests carries the risk of yielding a significant outcome by chance. Given that the data were normalized (though see comment #2 above), why not instead conduct a two-way RMANOVA (treatment group as between-subjects variable, and region of interest as the repeated measures/within-subjects variable)?

Response to comment 3

Thank you for your helpful suggestions. To analyze c-Fos counting, we referred to several reports (Ref. 25-27), which analyzed c-Fos positive neurons in several brain regions using unpaired t-test but not two-way ANOVA. We think it reasonable to use unpaired t-test in our study because we wanted to compare the average difference between saline treated mice and CNO treated mice. Moreover, we could not apply a two-way repeated ANOVA because the sample size and samples differed among brain regions. We added the description on the method used to analyze c-Fos counting to the Materials & Methods section (line 157).

Reviewer #1 comment 4

The manuscript refers to Table 1, but this information was not provided.

Response to comment 4

We apologize for this miss. Table 1 is inserted on page 40 and can be downloaded.

Reviewer #1 comment Minor issues

a) page 6, line 153: the statement that "...dorsal, but not ventral Hippocampus is involved in cognitive function..." is not well supported by the literature. For example see: Jin & Maren (2015) Scientific Reports; Okuyama et al. (2016) Science; Contreras et al. (2018) Hippocampus; Rao et al, (2019) Cell Reports; Jimenez et al. (2020) Nature Communications; to name a few example references.

b) the manuscript contains an overwhelming number of acronyms (OXT, ORM, SuM, PVN, ABC, CNO, NORT, AD, etc.), which makes some of the sentences challenging to understand. This reviewer suggests limiting the acronyms to a minimal number of terms that are used throughout.

c) page 10, line 249-252: This sentence is quite difficult to understand. Please revise.

d) page 11, line 267: "competed" should be revised to "compared"

e) page 13, line 314-315: The statement "...our result suggested that OXTergic neuron activity may be involved in each stage of AD pathological progression." is not supported by the results described in the current manuscript.

Response to comment Minor issues

(a) Thank you for recommending the given references. Accordingly, we added some new references to the manuscript. We would like to count c-Fos positive neurons following NORT. Therefore, we investigated the interaction between the hippocampus and object recognition memory. Several reports showed that the dorsal hippocampus was involved in object recognition memory (Ref. 21-24). We added relevant text to the Materials & Methods (line 160)

(b) ORM was removed.

(c) We revised this sentence in the Discussion section (Line 314).

(d) We revised this word.

(e) Thank you for your helpful suggestions. Indeed, in this study we did not perform any experiments related to Alzheimer’s disease. However, the present results show that the activity of OXTergic neurons regulates cognitive function. Further studies are required showing that OXTergic neurons are involved in the pathological progression of Alzheimer’s disease. We consider that this possibility is one of the future impacts of this study.

 

Responses to the comments raised by Reviewer #2

Reviewer #2 comment 1

As stated by the author in the introduction, "Projections of oxytocin (OXT) neurons exhibit extensive distribution across regions associated with cognitive functions. For instance, prior studies have revealed that OXT neurons project to the hippocampus (Hip), perirhinal cortex, entorhinal cortex (Ent), and supramammillary nucleus (SuM)." However, the findings from immunohistochemical staining (Fig. 1) merely illustrate the fibers within the paraventricular nucleus (PVN) and supramammillary nucleus (SuM). It would be advantageous to incorporate outcomes from other cerebral regions, notably the hippocampus (DG).

Response to comment 1

Thank you for your helpful suggestions. We have now added the missing information on OXT neurons projecting to the hippocampus. However, we could not detect the mCherry positive fiber in the hippocampus, including hippocampal DG, in this study. In a previous study, alkaline phosphatase staining did not detect OXT neurons projection in the hippocampus of Oxt cre/+; Z/AP mice [29], which might be because of the projection areas of OXT neurons detected depend on mice and staining methods. We have added relevant text to the Discussion section (line 304).

Reviewer #2 comment 2

Regarding c-Fos staining and quantification subsequent to CNO treatment, it is advisable for the authors to undertake additional verification to ascertain that the identified c-Fos positive neurons post-treatment indeed are OXTergic neurons.

Response to comment 2

Thank you for your pertinent suggestions. To ascertain that the identified c-Fos positive neurons post-treatment were indeed OXTergic neurons, we detected c-Fos and OXT by immunofluorescence staining. We detected double-labeled neurons (c-Fos+ and OXT+) in the PVN of CNO-treated mice. These results suggested that the c-Fos positive neurons observed post-treatment are OXTergic neurons. Therefore, CNO administration can activate OXT neurons. These results were added to supplement figure 3 and commented on in the Discussion section (line 309).

Reviewer #2 comment 3

Does the localized administration of CNO to the SuM have an impact on the DG? While the results section does not explicitly address the SuM-DG pathway, it extensively delves into this topic in the discussion. It would be better for the authors to substantiate their conceptual arguments in the discussion by presenting relevant empirical findings.

Response to comment 3

According to the reviewers’ suggestions, we have added the required description to the Discussion section (line 351). 

“Li et al. injected AAV-DIO-hM3Dq-mCherry or AAV-DIO-hM4Di-mCherry into the SuM in mice. CNO treatment increased c-Fos positive neurons in the DG of hM3Dq expressing mice but decreased c-Fos positive neurons in the DG of hM4Di expressing mice. In addition, Li et al. injected AAV-DIO-Ch2R-mCherry and AAV-shVglut2 (a marker of glutamatergic neurons) into the SuM in mice. Activating DG with optogenetics decreased c-Fos positive neurons in the DG of Vglut2 deletion mice. Accordingly, these results suggested that glutamatergic neurons in the SuM activate the DG (Ref. 9)”.

Reviewer #2 comment 4

In the discussion section (page 11, line 270), the authors put forth the assertion that “Thus, we propose that activating DG and SuM neurons enhance ORM by activating OXT neurons.” It would be judicious for the authors to complement this proposal by providing evidential support. Specifically, they could present data that demonstrate the comparable effects observed in “Effects of activation of OXTergic neurons in SuM by NORT”, where local activation of oxytocinergic fibers in the DG produced similar outcomes.

Response to comment 4

We appreciate your attempt to understand our discussion. However, we consider that it cannot activate OXTergic neurons in the DG because we could not detect OXT neuron projections in the hippocampus. Therefore, we proposed that OXTergic neurons in SuM projecting from PVN could enhance ORM via the activating of gluterergic neurons in DG projecting from SuM. We have added these comments to the Discussion section (line 367).

Reviewer #2 comment 5

In this study, only male mice were used as the subjects. Could potential distinctions arise in female mice?

Response to comment 5

Thank you for your helpful suggestions. Female mice may exhibit differences since the distribution of OXT receptors and OXT projection areas differ between female and male mice (Ref. 2). However, we did not use female Oxt-iCre mice to avoid the possibility of recombination in all cells in a Cre expressing fertilized egg. Moreover, DAT-Cre KI mice show sex-dependent changes in amphetamine-induced locomotor activity (Ref. 37). This report suggested that the phenotypes of Cre-KI mice were sex-dependent. Thus, we used male Oxt-iCre mice in this study. We have added these comments to the Discussion section (line 396).

---

## [Decision Letter · Decision Letter 1]

25 Oct 2023

Oxytocinergic Projection from the Hypothalamus to Supramammillary Nucleus Drives Recognition Memory in Mice

PONE-D-23-22233R1

Dear Dr. Saitoh,

We’re pleased to inform you that your manuscript has been judged scientifically suitable for publication and will be formally accepted for publication once it meets all outstanding technical requirements.

Kind regards,

Peng Zhong, Ph.D.

Academic Editor

PLOS ONE

Additional Editor Comments (optional):

Please make minor revision based on Reviewer 1's comments.

Reviewers' comments:

Reviewer's Responses to Questions

**Comments to the Author**

1. If the authors have adequately addressed your comments raised in a previous round of review and you feel that this manuscript is now acceptable for publication, you may indicate that here to bypass the “Comments to the Author” section, enter your conflict of interest statement in the “Confidential to Editor” section, and submit your "Accept" recommendation.

Reviewer #1: (No Response)

Reviewer #2: All comments have been addressed

2. Is the manuscript technically sound, and do the data support the conclusions?

Reviewer #1: Yes

Reviewer #2: (No Response)

3. Has the statistical analysis been performed appropriately and rigorously? 

Reviewer #1: Yes

Reviewer #2: (No Response)

4. Have the authors made all data underlying the findings in their manuscript fully available?

Reviewer #1: Yes

Reviewer #2: (No Response)

5. Is the manuscript presented in an intelligible fashion and written in standard English?

Reviewer #1: Yes

Reviewer #2: (No Response)

6. Review Comments to the Author

Reviewer #1: The revised manuscript is improved with the changes that the authors have made. A few issues remain to be addressed.

1. Measures of effect size should be included for each significant statistical value.

2. In response to the critique that the NORT results would be better analyzed as measures of discrimination ratio (difference score divided by total object exploration) instead of preference ratio. The authors seem to have disregarded this message and have now included difference score measures in the supplementary material. Both preference ratio and difference score measures suffer from the issue of high variability between groups and between individual mice. That is the reason for the original message that discrimination ratio measures represent a more rigorous measure of object memory in the NORT task. The discrimination ratio scores have a wider range of possible scores and generally have a reduced variability. Thus, analyses of NORT discrimination ratios tends to lead to fewer false interpretations of results.

3. The c-Fos counts indicate no differences in any of the examined areas after NORT between mice receiving saline or CNO. This result is a bit surprising given several reports of increased c-Fos in hippocampal cell layers and related regions 90 min after the training or test session of the NORT task (see Beer et al.; Mendez et al 2015a, 2015b; Cinalli et al. 2020; Bernstein et al., 2019; Barbosa et al., 2013, etc.). Given those reports, one might have expected differential counts between cingulate and CA1 and CA3, for example. As well differences might have been expected in the CNO treated mice if the drug was expected to have indirectly increased activity of hippocampal neurons. The authors should provide more interpretative information regarding these issues.

Reviewer #2: (No Response)

7. PLOS authors have the option to publish the peer review history of their article (what does this mean?). If published, this will include your full peer review and any attached files.

Reviewer #1: **Yes: **Robert Stackman Jr.

Reviewer #2: No

---

## [Editor Report · Acceptance letter]

7 Nov 2023

PONE-D-23-22233R1 

Oxytocinergic Projection from the Hypothalamus to Supramammillary Nucleus Drives Recognition Memory in Mice 

Dear Dr. Saitoh:

I'm pleased to inform you that your manuscript has been deemed suitable for publication in PLOS ONE. Congratulations! Your manuscript is now with our production department. 

Kind regards, 

on behalf of

Dr. Peng Zhong 

Academic Editor

PLOS ONE